# Dielectric Response of Different Alcohols in Water-Rich Binary Mixtures from THz Ellipsometry

**DOI:** 10.3390/ijms25084240

**Published:** 2024-04-11

**Authors:** Zahra Mazaheri, Gian Paolo Papari, Antonello Andreone

**Affiliations:** 1Department of Physics “E. Pancini”, University of Naples “Federico II”, Complesso MSA, 80126 Naples, Italy; zahra.mazaheri@unina.it (Z.M.); gianpaolo.papari@unina.it (G.P.P.); 2Naples Research Unit, National Institute for Nuclear Physics (INFN), Complesso MSA, 80126 Naples, Italy

**Keywords:** time-domain ellipsometry, binary water mixtures, molecular dynamics

## Abstract

We report a study on the hydrogen bonding mechanisms of three aliphatic alcohols (2-propanol, methanol, and ethanol) and one diol (ethylene glycol) in water solution using a time-domain ellipsometer in the THz region. The dielectric response of the pure liquids is nicely modeled by the generalized Debye–Lorentz equation. For binary mixtures, we analyze the data using a modified effective Debye model, which considers H-bond rupture and reformation dynamics and the motion of the alkyl chains and of the OH groups. We focus on the properties of the water-rich region, finding anomalous behavior in the absorption properties at very low solute molar concentrations. These results, first observed in the THz region, are in line with previous findings from different experiments and can be explained by taking into account the amphiphilic nature of the alcohol molecules.

## 1. Introduction

Recently, water-based mixture dynamics have attracted a great deal of interest in many different scientific fields for physical, chemical, engineering, and biological applications [1,2,3,4]. Despite this increased attention, the interaction between water and solute molecules, especially in the presence of a solute with amphiphilic behavior, is yet to be fully understood. Both hydrophilic and hydrophobic groups of an amphiphilic solute play a significant role in the intermolecular dynamics of the aqueous solution, leading to complex behavior in their mixtures. For instance, when dealing with alcohol molecules in the water-rich region, one has to take into account the effect of apolar or hydrophobic hydration. Many spectroscopic and thermodynamic measurements in alcohol–water binary mixtures with a low molar concentration of the solute strongly indicate the formation of low-entropy structures or “cages” of relatively common and long-lasting H bonds that are located around hydrophobic groups, often leading to an anomalous response [5]. A widely accepted explanation for the observed behavior is that each single molecule of alcohol is surrounded by water molecules at low concentrations [6]. An understanding of water-rich mixture dynamics is important not only for solution chemistry [7,8], but also for biosensing, since they indicate the activation parameters of organic reactions like the formation of micelles and biological membranes [9] or the conformational stability of proteins and nucleic acids [10].

Many spectroscopic methods have been used to investigate the unusual behavior of such mixtures such as neutron diffraction [11], nuclear magnetic resonance [12], Raman [13] and infrared spectroscopy [14], and recently time-domain THz spectroscopy [6] and ellipsometry [15], to name a few. In particular, the last two experimental techniques can provide useful information on the energy transitions and dynamics of aqueous mixtures, directly probing the H-bond mechanisms. The rotational and vibrational energies of water and many alcohol molecules correspond in fact to the frequency range between 100 GHz and 5 THz [16,17], which makes THz spectroscopic techniques a powerful probe for processes associated with hydrogen bond formation and rupture. The dielectric response of a polar liquid is the result of re-orientational motions of its molecules excited by an external field. Although the main fingerprints in the frequency spectrum lie in the microwave region, the relaxation events associated with hydrogen bond formation are on the order of ps and can therefore be probed in the THz regime.

Standard THz time-domain spectroscopy (TDS), however, in spite of being a coherent detection technique, does not offer enough precision and sensitivity for probing liquids, both in the transmission and reflection configuration. First, TDS needs a reference for an accurate determination of the sample parameters [18]. Moreover, this technique typically involves the use of a cuvette (with quartz or polymer windows) in the characterization process, which drastically reduces the signal-to-noise ratio and at the same time increases the uncertainty produced by the possible inaccuracy in sample and window thickness values.

Conversely, THz time-domain spectroscopic ellipsometry (TDSE) is a reference-free technique that allows for accurate sample characterization even in a single-shot mode. Furthermore, in the horizontal configuration we developed, it shows a number of practical advantages with respect to standard spectroscopy, such as the lack of need for a cuvette, no thickness limitations, and the absence of internal reflections, which make it a powerful and at the same time simple probe of pure liquids and mixtures in this frequency region [19].

In the following, using a time-domain ellipsometer, we report an accurate study on the H-bonding mechanisms of aliphatic alcohols differing in the length of the carbon chain: 2-propanol (isopropyl alcohol), methanol (methyl alcohol), and ethanol (ethyl alcohol). We also probe ethane-1,2-diol (ethylene glycol) to investigate whether the number of hydroxyl groups enhances H-bond formation in the solution [20,21].

Most of the recent work in the literature investigated the molecular dynamics of aqueous mixtures that span different ranges of the solute molar fraction. However, a specific focus on the water-rich region, where we believe one can better understand the subtle interplay between hydrophobic and hydrophilic mechanisms, is still lacking. Therefore, we first analyze the dielectric response of pure liquids and then the corresponding aqueous mixtures, focusing on the properties of the water-rich region. For very low solute molar concentrations (below 5%), we observe an anomalous behavior in the absorption properties in the THz region that can be ascribed to the competition between the hydrophobic and hydrophilic behavior of alcohol molecules.

## 2. Results and Discussion

The well-known phenomenological prediction of the dielectric response of polar liquids such as alcohols is based on the Debye model, in which the main role in the liquid dynamics is played by the time-dependent response to a specific external excitation, described using the relaxation time τ. The number of relaxation events in a pure polar liquid is based on the mesoscopic structure of its molecules. When several relaxation events are present, in order to accurately describe the complex dielectric response of the liquid, the generalized Debye Equation [22] is used as follows:(1)$${\tilde{\epsilon }}_{D}\left(\omega \right)=\epsilon_{\infty }+\sum_{i=1}^{N}\frac{\epsilon_{i}-\epsilon_{i+1}}{1+i\omega \tau_{i}},$$
where $\epsilon_{\infty }$ is the infinite dielectric constant and *N* is the number of relaxation events. The dielectric strength $\epsilon_{i}-\epsilon_{i+1}$ and the constant time $\tau_{i}$ are associated with the *i*-th event.

Alcohols have three relaxation events: a slow mechanism due to molecule rotational movement ($\tau_{1}$), followed by an intermediate and a faster process, corresponding to the switching of H-bonds at the end ($\tau_{2}$) and within each chain ($\tau_{3}$) [23], which can be all effectively described using the above equation. However, especially when dealing with highly polar liquids, the Debye model at very high frequencies may fail in accurately describing the relaxation dynamics, since in such cases, the disruption of some molecules affects the surrounding molecules and in return produces (Lorentz) resonance. In fact, for polar liquids, the dielectric response is determined by dielectric relaxation processes at lower energies and by vibrational (intra- and inter-molecular) modes at higher energies [24]. For instance, water has an intermolecular vibration mode at 5 THz [25]. In this regard, to fully investigate the complex dynamics of alcohol molecules in the THz region, one may need to take into account the effects of both low-frequency (relaxation events) and high-frequency (molecule vibrations) dynamics. Even though the latter mechanism is usually not really effective in the low THz regime, depending on the frequency window and the polarity of the liquid, one or more Lorentzian terms describing the contribution of the resonance peaks in the frequency spectrum might improve the Debye prediction. This can be achieved by introducing a modified equation, where at least one Lorentzian term is added to the generalized Debye model:(2)$$\tilde{\epsilon }\left(\omega \right)={\tilde{\epsilon }}_{D}\left(\omega \right)+\frac{A_{1}}{\omega_{1}^{2}-\omega^{2}-i\omega \gamma_{1}},$$
where $\gamma_{1}$ is the linewidth and $A_{1}$ is the amplitude of the resonance at $\omega_{1}$ [26].

Fitting the real and imaginary parts of the experimental dielectric function to the aforementioned model, we can retrieve the Debye parameters of the pure alcohols.

The Debye model describes the frequency response of a pure liquid well, where different relaxation events govern the molecular dynamics, based on the assumption that these processes are independent of each other and take place in a parallel fashion. Moving on to aqueous binary mixtures, this assumption is not applicable, since molecules in the water solution will start to interact; therefore, a deeper insight into the fundamental nature of the relaxation processes is required. Many attempts have been made over the years to model their molecular dynamics by extending the Debye theory so that one can simply predict the dielectric response of an ideally interacting mixture starting from the dielectric response of the pure components. Indeed, in comparison with other binary solutions, such as alcohol–alcohol [27], water–alcohol solutions have been always considered as complex systems since they exhibit a larger non-ideal response, which makes the study of their behavior definitely more challenging. Moreover, because of the amphiphilic nature of alcohol and diol molecules, such a mixture behaves differently at various concentrations of the solute. In a previous work [15], we have already shown that when increasing the alcohol concentration in the mixture, the effective Debye model (describing fully interacting liquids) fails in predicting the observed dielectric response because inhomogeneous mixing affects the long-range collective H-bonding dynamics of water. This gives rise to a reduction in the rotational movements of molecules and therefore in the value of $\tau_{1}$. Very recently, this slower relaxation time has been ascribed to an increase in the potential energy barrier, which in turn limits the activation energy for water molecule rotation [28].

An effective Debye relaxation model for mixed solutions was first presented by Lou et al. [29]. Recently, Zhou and Arbab [22] tried to predict the dynamics of binary water–alcohol solutions in the THz region by introducing a three-term model. To describe the complex dielectric function of two different polar liquids in the mixture, the parameters of Equation (Equation 1) were replaced using empirical rules (for dielectric strengths, this is the symmetric Bruggeman rule [30]), and each contribution of the solvent and the solute was weighted in terms of the molar fraction $X_{M}$, defined as the ratio between the moles of the solute and the total moles of the solution.

According to [22], the main equations that govern a mixture of water and alcohol, assuming that they behave as interacting liquids, are:(3)$$\frac{\left(1-X_{M}\right)\left(\epsilon_{i,w}-\epsilon_{i,eff}\right)}{\epsilon_{i,w}+2\epsilon_{i,eff}}+\frac{X_{M}\left(\epsilon_{i,a}-\epsilon_{i,eff}\right)}{\epsilon_{i,a}+2\epsilon_{i,eff}}=0$$
and
(4)$$ln\left(\tau_{i,eff}\right)=\;X_{M}ln\left(\tau_{i,a}\right)+\left(1-X_{M}\right)\;ln\left(\tau_{i,w}\right)$$
where subscripts *a* and *w* refer to alcohol and water, respectively, and $\epsilon_{i,eff}$ is the effective dielectric constant in the mixture associated with the *i*-th event. Since water molecular dynamics is described using only two relaxation events, then *i* = 1 or 3. We then empirically assume that $\tau_{2,eff}=\tau_{2,a}/X_{M}$ and $\epsilon_{2,eff}=\epsilon_{2,a}$.

We first analyzed the dielectric response of the pure aliphatic alcohols and the diol. Figure 1 presents the frequency-dependent dielectric properties of (a) 2-propanol, (b) methanol, (c) ethanol and (d) ethane-1,2-diol in the THz region determined by the TDSE technique and the corresponding fitting curves (dashed lines).

The experimental results confirm previous findings [31] that as the alkyl chain in the alcohol increases (ethanol, 2-propanol), both the refractive index and extinction coefficient decrease. Alcohols with a longer alkyl chain are, in fact, less polar, and the polarity of the liquid has a direct relation to its absorption properties [32]. The fit procedure is based on a nonlinear regression method, simultaneously applied to Re($\tilde{\epsilon }$) and Im($\tilde{\epsilon }$). All dielectric strengths $\epsilon_{i}$ are kept fixed to the values reported in the literature [15,22,27,33], while $\epsilon_{\infty }$ is set considering the value extrapolated from the experimental data. Relaxation times $\tau_{2}$ and $\tau_{3}$ are only considered free parameters. In alcohols in fact, the slowest relaxation time $\tau_{1}$ is on the order of hundreds of picoseconds and lies consequently in a range not accessible to the THz probe; therefore, its value was taken from previous experiments [22,33,34]. The parameters used in the minimization procedure are summarized in Table 1. The results for pure water are also shown, which nicely match previous data taken in a frequency and temperature range similar to each other [22,35]. The Debye model with three relaxation events (Equation (Equation 1)) is used to describe the behavior of all alcohols except ethane-1,2-diol, where for the observed frequency dependence (see Figure 1d), Equation (Equation 2) with Lorentzian resonance is more fitting. For this case alone, the frequency $\omega_{1}$, the line width $\gamma_{1}$, and the amplitude $A_{1}$ of the ethane-1,2-diol resonance are also considered variable terms in the nonlinear fit algorithm, yielding $\omega_{1}$ = 3.9±0.1 THz, $\gamma_{1}=2.2\pm 0.1$ THz, and $A_{1}=0.010\pm 0.003$
${\mathrm{THz}}^{2}$. Indeed, pure aliphatic alcohols do exhibit a resonance peak between 1 and 2 THz [17]. However, as reported earlier [34], the presence of resonant contributions in the THz region in standard alcohols cannot be easily observed from the experimental data due to their low intensity and the dominance of the upper frequency tail of the dielectric relaxation processes that occur in the microwave region. Moreover, these vibrational modes show damping term values $\gamma_{1}$ comparable to center frequency values $\omega_{1}$, which ’wipe out’ the peaks.

The experimental results and corresponding fits plotted in Figure 1 demonstrate that the generalized Debye model properly describes the dielectric response of pure alcohols. Therefore, the parameters listed in Table 1 can be safely used to check the validity of the Debye relaxation model for mixed solutions.

In Figure 2, we show the extinction coefficient determined by ellipsometric measurements, averaged using four different frequencies (0.3,0.6,0.9 and 1.2 THz), as a function of the molar fraction $X_{M}$ for all water–alcohol and water–diol solutions under investigation. According to a simple mixing rule, by adding a component with a value of *k* lower than water, a monotonous decrease in the effective extinction coefficient is expected, since the water molecules are simply replaced by the less absorbing alcohol molecules. This effect is often observed in hydrophobic solutes [21]. In reality, due to the amphiphilic behavior of alcohols, their interaction with water molecules is more complex and clearly affects the dielectric response of the solution in the water-rich region. In fact, for very low values of the solute fraction, the results strongly differ from the prediction given by the effective Debye model, described by the dot-dashed curve in each graph of Figure 2.

The results shown in Figure 2 are consistent with previous measurements, mainly of thermodynamic nature. In previous studies, the larger than expected decrease in entropy and enthalpy due to mixing in aqueous solutions has been ascribed to an improvement in the structure of water molecules in the hydration shell of the solute molecule [36], resulting in an increase in the structure similar to that of ice [37]. However, recent findings indicate that the so-called ’iceberg model’ is not capable of describing, in the whole range of concentrations, the behavior of water-based binary mixtures. Based on the change in concentration of the solute, a number of anomalies have been observed in different works [21,22,23,35,38,39]. Additionally, the isomers act differently when mixed with water [20,40]. The positions of the hydroxyl groups and carbon chain structures differ from one alcohol to the other, affecting the formation of hydrogen bonds in the different solutions [32].

To discuss the molecular dynamics of the mixture at very low $X_{M}$, we show in Figure 3 the deviation in the experimental results from the expected Debye behavior ($\Delta k=k_{exp}-k_{Debye}$). This plot better highlights that at a low alcohol molar fraction (0–5%), the water mixtures behave completely differently from the Debye prediction.

Three molar regions can roughly be distinguished, starting from pure water ($X_{M}=0\%$) and delimited by $X_{M_{1}}$ and $X_{M_{2}}$. These are nominal values only, which might change depending on the alcohol. In the region below $X_{M_{1}}$ (represented by the dark shaded area in Figure 3), the presence of hydrophobic groups in the very scarce alcohol molecules rules the intermolecular dynamics, so that the existing H bonds of the bulk water are disturbed, leading to a sudden decrease in absorption properties [41].

In the range between $X_{M_{1}}$ and $X_{M_{2}}$ (light-shaded area in the same figure), there is competition between hydrophobic and hydrophilic mechanisms, with the latter prevailing at the end: water surrounds alcohol molecules, trying to form new H-bonds and leading to the formation of clusters that “boost” the absorption properties of bulk water. This results in a sharper increase in *k* for all aliphatic alcohols and a milder increase for ethylene glycol only.

Above $X_{M_{2}}$, the growth in H-bond formation between host and guest molecules leads to a behavior closer to the Debye-like response of the binary mixture.

For all alcohols except ethanol, an interaction between rupture and formation of H bonds is observed below a 5% molar fraction. There seems to be a difference, however, in the extension of the “competing region”. We observe that the length of the alcohol chain, especially the hydrophobic group size, plays an important role in the intermolecular dynamics of the alcohol solution. The longer the hydrophobic chain, the larger the effect of hydrophobic hydration, which in turn leads to the observation of a maximum deviation in the expected behavior of the mixture corresponding to very low molar fractions [36]. This is clearly seen in the case of 2-propanol, which has the longest hydrophobic chain, which leads to lower values of both $X_{M_{1}}$ and $X_{M_{2}}$ than the other mixtures under test.

Ethanol is a notable exception, since it forms weaker hydrogen bonds in solution [42], and therefore it overall deviates the most and for larger molar fractions from the effective Debye prediction. In contrast, it is interesting to observe that the insertion of ethane-1,2-diol in water shows only a slight deviation from the expected behavior within the competing region. We speculate that this is due to the presence of a double hydroxyl group, which helps the intermolecular interaction and determines the diol’s response in the mixture more than its short (hydrophobic) chain does.

## 3. Materials and Methods

The time-domain ellipsometer used for the measurements is based on a femtosecond laser, fiber-coupled to two photoconductive antennas used for the emission and detection of the pulsed electric field in a coherent fashion. We developed a highly flexible custom system with an in-house calibration routine that allows us to probe the polarized response of solids and liquids with ease. Compared to a previous setup [15], the region used for the spectral investigation presently ranges from 0.1 THz to 1.6 THz, spanning more than a frequency decade and improving the bandwidth by a factor of 2. Polimeric TPX (polymethylpentene) lenses and conducting wire grid polarizers (WGPs) are used to collimate and focus the THz pulse and to control and select the polarization state of the signal, respectively. More details on the measurement setup can be found in [19].

We investigated 2-propanol, from ROMIL Ltd. (Cambridge, UK), and methanol, ethanol and ethane-1,2-diol, from Merck-Aldrich (St. Louis, MO, USA), all with a purity higher than 99.7%, and compared their response with that of deionized Milli-Q water. Binary mixtures were realized by gradually adding the solute to pure water with a precision of 0.05 mL. In this way, we prepared sets of aqueous samples with different alcohol molar fractions, with a focus on the water-rich region starting from a concentration of 0.04%. All measurements were carried out in a controlled environment with the temperature set at 25±1 °C.

In Figure 4, a pictorial representation of the experimental setup is shown schematically. From the measurement of the time-dependent electric field of the reflected signals (s- and p-polarized components), direct information on the ellipsometric parameters Ψ and Δ, defined as the amplitude ratio and the phase difference of the states s and p, respectively, as a function of frequency [43]. Ψ and Δ are linked to the complex optical response $\tilde{n}\left(\omega \right)=n\left(\omega \right)-ik\left(\omega \right)$ of the sample under investigation via the following equations [44]:(5)$$n^{2}-k^{2}=\sin^{2}\theta \left[1+\frac{\tan^{2}\theta \left(\cos^{2}2\Psi -\sin^{2}2\Psi \sin^{2}\Delta \right)}{{\left(1+\sin2\Psi \cos\Delta \right)}^{2}}\right]$$
(6)$$2nk=\sin^{2}\theta \frac{\tan^{2}\theta \sin4\Psi \sin\Delta }{{\left(1+\sin2\Psi \cos\Delta \right)}^{2}}.$$
where θ is the incident angle of the beam and *n* and *k* represent the refractive index and the extinction coefficient, respectively.

To improve the precision in evaluating the optical response, θ is set at 55 °, close to the pseudo-Brewster angle $\theta_{B^{\prime }}$ of all alcohols investigated (for isopropyl alcohol, for example, $\theta_{B^{\prime }}$≈ 58 ° in this frequency range). In this way, in fact, the incident angle chosen provides the lowest possible errors in the evaluation of the ellipsometric angles (Ψ; Δ) [43]. This translates to a maximum relative error in the retrieved values of the material properties, *n* and *k*, of around 2% and 5%, respectively.

## 4. Conclusions

In this work, the high-frequency behavior of a binary solution consisting of Milli-Q water, various alcohols, and a diol with different molar concentrations was measured using a THz TDSE technique. The customized setup we built is capable of detecting small changes in the optical response of pure liquid water due to the introduction of a small amount of solute (down to a 0.04% molar fraction for alcohols) and capable of probing the molecular dynamics of the mixture using a single measurement.

First, the dielectric response of the pure alcohols is analyzed and nicely fitted by using a generalized Debye relaxation model, which allows us to retrieve accurate information on the relaxation events occurring in each liquid in the terahertz frequency domain. In the case of ethane-1,2-diol only, due to the presence of a resonance in the vicinity of 4 THz, the Debye model is modified to also take into account the Lorentz term due to vibrational modes.

Then, based on the Debye parameters extracted from the pure constituents, the optical properties of aqueous binary mixtures upon increasing the alcohol molar fraction are measured and compared with the theoretical prediction based on an effective Debye model for interacting liquids. Our results, and their disagreement with the behavior expected using classical mean-field theory, shed light on the complex dynamics occurring in aqueous binary mixtures due to the competing hydrophobic and hydrophilic behavior of alcohols. In the water-rich region (with very scarce solute molecules), we measure a sudden drop in the absorption properties, with the dominant mechanism appearing to be the destruction of existing H-bonds between water molecules, since the hydrophobic behavior of the alcohol plays the main role. As the solute molar concentrations become larger, we first observe an increase in the absorption properties of the mixture because of the formation of water clusters around alcohol molecules, followed by a Debye-like decay when the alcohol concentration. These results can be used to improve our understanding of the amphiphilic nature of alcohol molecules in organic liquids. Hydrophobicity and hydrophilicity are key ingredients in water solutions, and their mechanisms play a pivotal role in different physical and biological processes.

## Figures and Tables

**Figure 1 ijms-25-04240-f001:**
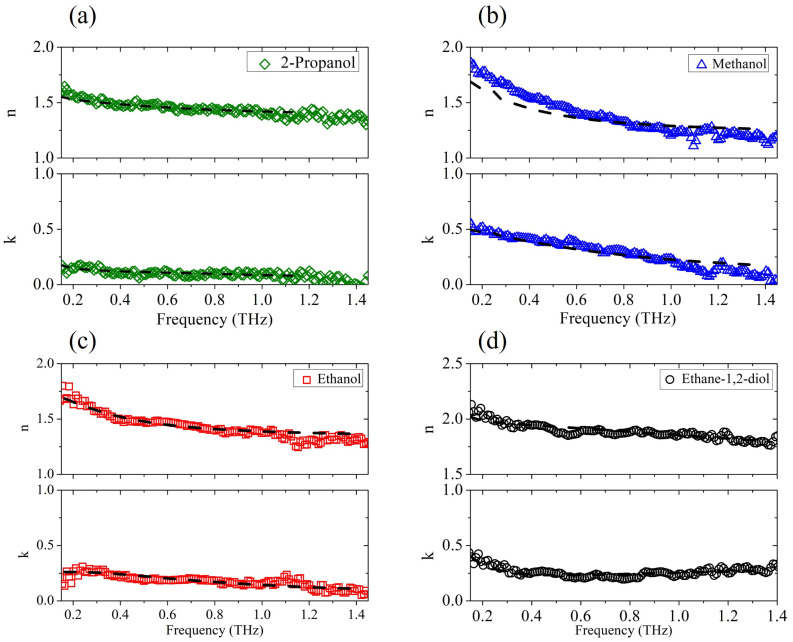
The experimental complex refractive index n−ik vs. frequency (solid points) for (**a**) 2-propanol, (**b**) methanol, (**c**) ethanol, and (**d**) ethane-1,2-diol, and its fit (dot-dashed lines) to the generalized Debye model.

**Figure 2 ijms-25-04240-f002:**
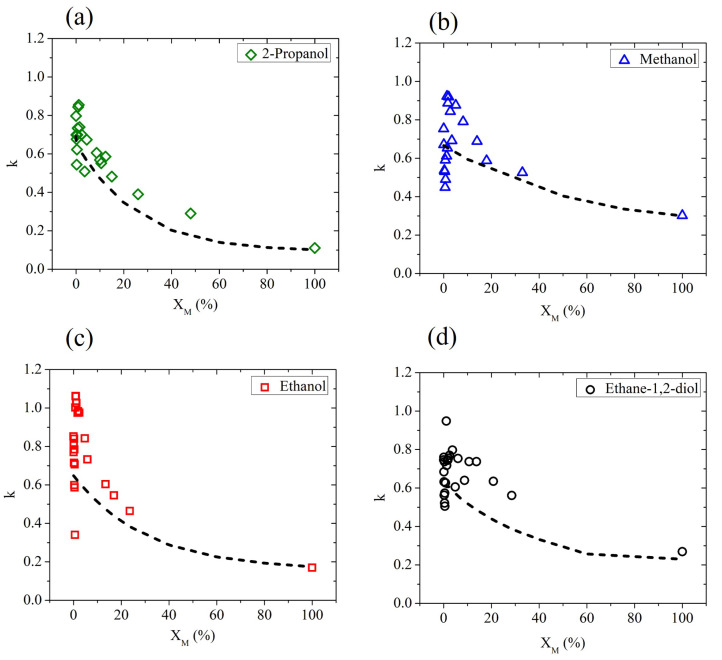
The experimental values (averaged over four different frequency points: 0.3,0.6,0.9, and 1.2 THz) of *k* in aqueous binary mixtures based on (**a**) 2-propanol, (**b**) methanol, (**c**) ethanol, and (**d**) ethane-1,2-diol measured as a function of the solute molar fraction (solid points). Dot-dashed lines describe the prediction of the effective Debye model.

**Figure 3 ijms-25-04240-f003:**
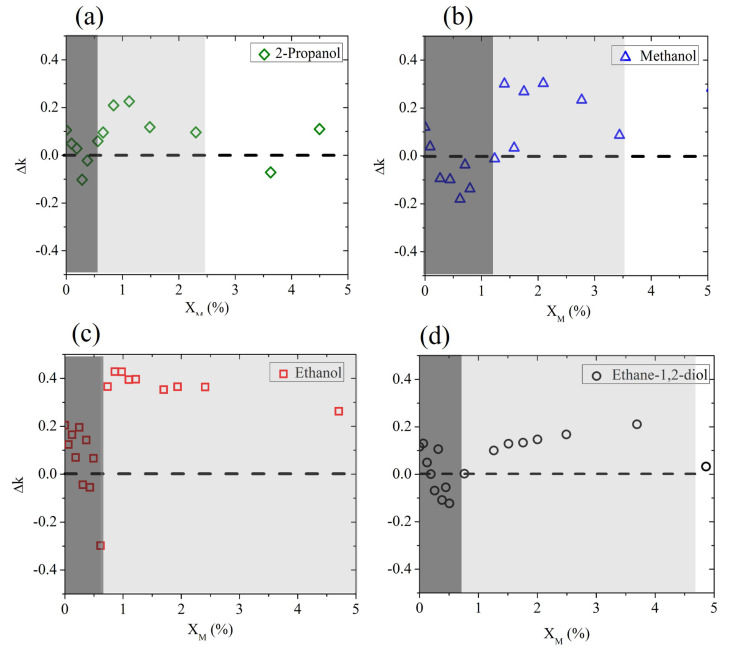
Deviation Δk (frequency averaged) between the experimental data and the prediction of the Debye model for aqueous binary mixtures based on (**a**) 2-propanol, (**b**) methanol, (**c**) ethanol, and (**d**) ethane-1,2-diol, measured at a low solute molar fraction. The dashed horizontal line marks the difference with the expected Debye behavior.

**Figure 4 ijms-25-04240-f004:**
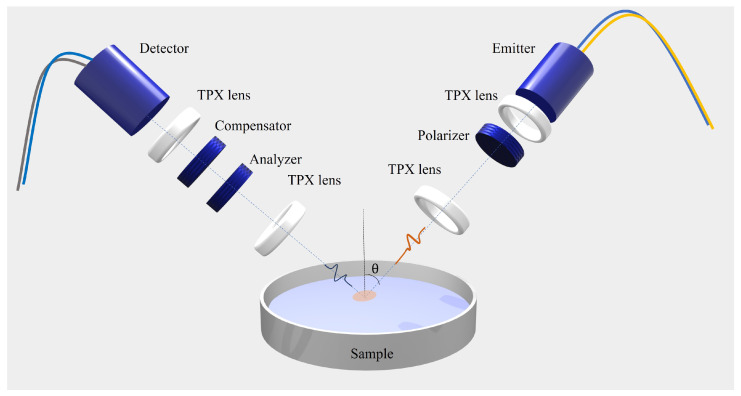
Sketch of the opto-mechanical setup depicting the customized fiber-coupled THz TDSE.

**Table 1 ijms-25-04240-t001:** Debye parameters of water, pure aliphatic alcohols, and ethane-1,2-diol.

Liquid	$\epsilon_{\infty }$	$\epsilon_{1}$	$\epsilon_{2}$	$\epsilon_{3}$	$\tau_{1}\left(\mathrm{ps}\right)$	$\tau_{2}\left(\mathrm{ps}\right)$	$\tau_{3}\left(\mathrm{ps}\right)$
Water	3.1	78.4	-	4	8.8 ± 0.5	-	0.20 ± 0.02
2-Propanol	2	19.4	4.8	3.9	351	4.8 ± 0.9	0.35 ± 0.04
Methanol	1.8	32.5	4.9	2.8	52	1.94 ± 0.04	0.51 ± 0.04
Ethanol	1.9	24.4	4.4	3	161	4.9 ± 0.8	0.50 ± 0.03
Ethane-1,2-diol	2.5	42	6.6	3.7	120	1.84 ± 0.02	0.10 ± 0.01

## Data Availability

The data presented in this study are available on request from the corresponding author.

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
