# Peer review of "Dielectric Response of Different Alcohols in Water-Rich Binary Mixtures from THz Ellipsometry"

_ijms, 2024, doi:10.3390/ijms25084240_

Round 1

Reviewer 1 Report

Comments and Suggestions for Authors

Review attached

Reviewer 2 Report

Comments and Suggestions for Authors

The manuscript "Dielectric Response of Different Alcohols in Water-rich Binary Mixtures from THz Ellipsometry" describes the experimental measurement of the complex refractive index n + ik vs frequency for four alcohols (methanol, 2-propanol, ethanol, ethylene glycol) as well as their mixtures with water at four weighted different frequency points. The applicability of Debye model for the systems under study is confirmed, in general. The smooth increase in influence of hydrophobic interactions over hydrophylic at low molar fractions of alcohols, where Debye model fits the experimental data the worst, is shown.

In general, the paper is good and sound, and only a couple minor questions should be asked:

1. I personally dislike the fact that on Fig. 3 the shape of curve is strongly determined by the most "right" points (corresponding to XM = 1). The gap between the end points and previous points should also be filled with data. An obtaining of a couple of additional points is in order.

2. Why in Table 1 some data (tau1) were borrowed from the literature? What would result in a calculation, where all three characteristic times were optimized? Would tau1 optimized agree with a literature value?

3. Could you assume why there is no clear regularity in changes of k dependence in the homologous series methanol->propanol-2?
